# A Time-Kill Assay Study on the Synergistic Bactericidal Activity of Pomegranate Rind Extract and Zn (II) against Methicillin-Resistant *Staphylococcus aureus* (MRSA), *Staphylococcus epidermidis,* *Escherichia coli*, and *Pseudomonas aeruginosa*

**DOI:** 10.3390/biom11121889

**Published:** 2021-12-16

**Authors:** Amal Alrashidi, Mohammed Jafar, Niamh Higgins, Ciara Mulligan, Carmine Varricchio, Ryan Moseley, Vildan Celiksoy, David M. J. Houston, Charles M. Heard

**Affiliations:** 1School of Pharmacy and Pharmaceutical Sciences, Cardiff University, Cardiff CF10 3NB, UK; alrashidia@cardiff.ac.uk (A.A.); jafarm@cardiff.ac.uk (M.J.); higginsn1@cardiff.ac.uk (N.H.); mulligancf@cardiff.ac.uk (C.M.); varricchioc@cardiff.ac.uk (C.V.); celiksoyv@cardiff.ac.uk (V.C.); houstondm@cardiff.ac.uk (D.M.J.H.); 2Oral and Biomedical Sciences, School of Dentistry, Cardiff University, Cardiff CF14 4XY, UK; moseleyr@cardiff.ac.uk

**Keywords:** PRE, zinc, synergistic activity, time-kill assay, *Staphylococcus aureus* (MRSA), *Staphylococcus epidermidis*, *Escherichia coli*, *Pseudomonas aeruginosa*

## Abstract

There is a need for new antimicrobial systems due to increased global resistance to current antimicrobials. Pomegranate rind extract (PRE) and Zn (II) ions both possess a level of antimicrobial activity and work has previously shown that PRE/Zn (II) in combination possesses synergistic activity against *Herpes simplex* virus and *Micrococcus luteus*. Here, we determined whether such synergistic activity extended to other, more pathogenic, bacteria. Reference strains of methicillin-resistant *Staphylococcus aureus* (MRSA), *Staphylococcus epidermidis*, *Escherichia coli*, and *Pseudomonas aeruginosa* were cultured and subjected to challenge by PRE, Zn (II), or PRE + Zn (II), in time-kill assays. Data were obtained independently by two researchers using different PRE preparations. Statistically significant synergistic activity for PRE + Zn (II) was shown for all four bacterial strains tested compared to untreated controls, although the extent of efficacy and timescales varied. Zn (II) exerted activity and at 1 h, it was not possible to distinguish with PRE + Zn (II) combination treatment in all cases. PRE alone showed low activity against all four bacteria. Reproducible synergistic bactericidal activity involving PRE and Zn (II) has been confirmed. Potential mechanisms are discussed. The development of a therapeutic system that possesses demonstrable antimicrobial activity is supported which lends itself particularly to topical delivery applications, for example MRSA infections.

## 1. Introduction

Antimicrobial resistance is a natural phenomenon, occurring as a consequence of gradual changes in bacterial gene expression to facilitate the development of resistance against certain antibiotic modalities. The current global antibiotic resistance crisis has been particularly attributed to the overuse and misuse of these therapeutics, in addition to other factors—such as inappropriate antibiotic prescribing, extensive agricultural use, and the lack of new antibiotic drug development—have all contributed to the dramatic emergence of modern drug-resistant pathogens [1,2]. With the recent dramatic increases in resistance to existing antimicrobial agents, certain infections have become extremely difficult to treat, with generally stricter regulations being applied to antibiotic administration. For example, methicillin-resistant *Staphylococcus aureus* (MRSA) infection is caused by *Staphylococcus aureus* that has developed resistance to many of the antibiotics used to treat such infections, which typically occur in people who have been in hospitals and are difficult to treat. Therefore, there is an urgent need to economically develop more novel agents and approaches to treat microbial infections. This is an important consideration, as the accessibility of affordable new medicines to middle-to-low-income countries has been recognised as a priority by World Health Organization (WHO), where the use of traditional and complementary medicines is emphasised, as the costs of developing new single chemical entities would be prohibitive [3]. Furthermore, there is a recognised predilection of the populous for medicines that are of natural origin.

The pomegranate, the fruit of *Punica granatum* L., has a longstanding history as a folklore medicine and in the treatment of various bacterial infections, which has been generally supported having been wide number of studies in recent times [4]. Its anti-microbial properties have been largely attributed to the polyphenolic contents, of which the hydrolysable ellagitannins are the dominant class [5]. Such phytochemicals are concentrated in the fruit exocarp (or ‘rind’) although other extracts of other parts of the fruit—such as the arils—also possess anti-microbial/bactericidal effects [6]. Al-Zoreky (2002) demonstrated that pomegranate peel exhibited bactericidal activity against *Listeria monocytogenes*, *Staphylococcus aureus*, *Escherichia coli*, *Salmonella enteritidis*, and *Yersinia enterocolitica* [7]. Similarly, Nozohour et al. (2018) demonstrated the activity of pomegranate rind extract (PRE) against methicillin-resistant *Staphylococcus aureus* (MRSA) and *Pseudomonas aeruginosa* [8]. Studies have shown that punicalagin (Figure 1) and its metabolites such as ellagic acid, punicalin and the gut metabolite urolithin A—possess a range of antioxidant, anti-inflammatory, and antimicrobial properties [9,10].

Although it is widely recognised that punicalagin is largely responsible for the bioactivity of PRE, there is a rationale for developing holistic pomegranate extracts as antimicrobials. It has been reported that >120 different phytochemicals are found in pomegranate extracts, and the significance of combination therapy in combating resistance mechanisms is of increasing interest [11,12,13]. Multidrug resistance can arise through several different resistance genes, each providing resistance to a particular antibiotic, or a single resistance mechanism giving resistance to more than one antibiotic [14]. For example, extended spectrum β lactamase (ESBL)-producing Gram-negative bacteria like *E. coli* and *Klebsiella pneumoniae*. ESBLs are enzymes that inhibit the activity of many clinically important antibiotics. Thus, infections with bacteria expressing ESBLs are hard to treat and are becoming increasingly common [15].

In another facet of exploring combination therapies, natural products have been combined with other substances to achieve enhanced or synergistic action, including as anti-microbials [16]. In particular, the enhancement of PRE activity has been explored with the co-application of transition metal ions, including Fe (II) and Cu (II) [17,18], and notably Zn (II), with significant synergistic (potentiated) activities having been found against the *Herpes simplex* virus (HSV-1, HSV-2 and acyclovir-resistant HSV-2) [19] and bacteria, such as *Micrococcus luteus* [20].

In an effort to further explore the broad-spectrum nature of the PRE/Zn (II) combinational therapy, the current work aimed to determine if the same general conditions that displayed synergistic bactericidal activity reported previously, particularly against the bacterium *M. luteus*, would also be observed against a panel of other bacteria of a more pathogenic nature. In this paper, we examined the time-kill effects of PRE, Zn (II) and PRE/Zn (II) combination against the Gram-positive bacteria, MRSA, and *Staphylococcus epidermidis* and the Gram-negative bacteria, *Escherichia coli*, and *Pseudomonas aeruginosa*.

## 2. Materials and Methods

### 2.1. Materials

Pomegranates were obtained from local supermarkets, and were of Spanish origin. Zn (II) as zinc sulphate heptahydrate (ZnSO_4_·7H_2_O), potassium hydrogen phthalate HPLC-grade methanol and HPLC-grade water were purchased from ThermoFisher Scientific (Loughborough, UK). Punicalagin (≥98%), ascorbic acid and sodium carbonate (Na_2_CO_3_) were purchased from Sigma-Aldrich (Gillingham, UK). Mueller–Hinton broth (MH broth) and Mueller–Hinton agar (MH agar) were both purchased from Oxoid (Basingstoke, UK).

### 2.2. PRE Preparation and Evaluation

PRE was produced by the hot aqueous extraction method, followed by freeze drying [19,20]. Pomegranates of Spanish origin were bought from a local supermarket and the rinds were excised then cut into strips prior to blending in deionized water 25% w/v. This was boiled for 10 min before being centrifuged for 40 min on 6 occasions using Heraeus™ Multifuge 3 S-R centrifuge at 5980× *g*. Next, the solution was vacuum filtered through 0.45 µm nylon membrane membrane. The total volume of the filtrate was 289.5 mL which was then freeze-dried using a Scanivac^TM^ freeze drier before being stored at −20 °C. The freeze-dried PRE was reconstituted with phthalate buffer (pH 4.5) as required.

Quantitative analysis of PRE and punicalagin by reverse-phase HPLC using an Agilent series 1100 HPLC system fitted with a Kinetex, 5 µm C18 100 Å 4.6 × 150 mm (Phenomenex, Macclesfield, UK) was used, along with a binary gradient elution programme involving A: methanol with 0.1% trifluoroacetic acid (TFA) and B: deionised water with 0.1% TFA (Table 1). Analyte detection was by UV at 258 nm and the analysis was performed at room temperature. Injection volume was 20 μL and the flow rate was 1.5 mL/min; the total run time was 30 min.

### 2.3. Microbiological Evaluations

Four bacteria were used for investigation used in this study: MRSA (NCTC 12493), *S. epidermidis* ATCC 14990 (NCTC 11047), *E. coli* (NCTC 12923), and *P. aeruginosa* (NCTC 6750). Checkboard analysis was performed using standard techniques [20]. Time-kill assays were performed independently by two researchers on all four bacteria, using two different preparations of PRE.

Solutions of test substance in 990 μL phthalate buffer (and negative control phthalate buffer only) were added to a 10 µL aliquot of 0.5 McFarland standard of tested bacteria in a 2 mL Eppendorf vial. After vortex mixing, incubation took place for different time points: 5, 10, and 20 min; and additionally 60 min for MRSA and *S. epidermidis.* After each time-point, 100 µL aliquots were transferred to 900 µL of neutralizing agent (universal quenching agent, UQA) which was added to all experimental groups (including negative control) to halt the antimicrobial action of compounds and their combination and vortex mixed. After 60 min, serial dilutions of the neutralized mixture were prepared out up to 8 times. Next, 3 drops in 10 µL volume of each dilution were transferred to an MHA plate and incubated under aerobic conditions at 37 °C for 24 h. The number of surviving colonies were then enumerated [21]; and the following formula used to calculate the colony forming units per one millilitre (CFU/mL): (no of colonies × dilution factor)/volume of culture media

The Log_10_ reduction in CFU/1 mL was determined versus control (phthalate buffer, pH 4.5) using the following Equation (1), where A is CFU/mL of the control (phthalate buffer) and B is CFU/mL of test sample.
〖log〗_10 (A) − 〖log〗_10 (B)(1)

### 2.4. Statistical Analysis

Experiments were performed by two researchers using two different PRE preparations, and each researcher repeated experiment nine times for each microbe. Then results from each researcher were collected and data presented as mean ± standard deviation (SD). GraphPad Prism 8.0 software (GraphPad Software, San Diego, CA, USA) was used for statistical analysis: one-way ANOVA was performed with post-test Tukey correction: *p* < 0.05 was considered statistically significant and in the graphical plots denoted as * *p* ≤ 0.05, ** *p* ≤ 0.01, *** *p* ≤ 0.001, and **** *p* ≤ 0.0001.

### 2.5. Computational Surface Charge Analysis

All molecular modelling experiments were performed on Asus WS X299 PRO Intel^®^ i9-10980XE CPU @ 3.00 GHz × 36 running Ubuntu 18.04. The molecular structures were prepared by MOE QuikPrep tool generating possible ionization states at pH 7.4 and pH 4.5. Quantitative calculation of formal charge and relative negative charges were calculated at both pH values (Molecular operating environment (MOE), Montreal, QC, Canada).

## 3. Results

### 3.1. PRE Evaluation

The chromatogram shown in Figure 1 highlights the multiple tannin compounds present in PRE and—in particular—punicalagin, which exists as two anomers in a characteristic 1:2 ratio: α-punicalagin with retention time of 7.38 min and anomer β-punicalagin with retention time 11.74 min. The identity of punicalagin was also confirmed of punicalagin by comparison with purified punicalagin. Spiking with standard punicalagin gave rise to both α and β punicalagin peaks, and this was also used to prepare a standard calibration curve from which it was determined that the mean punicalagin level in the 2 extracts was PRE was 21.7% (1 mg in each 4.61 mg of PRE). The level of punicalagin, which is known to be the main active component of PRE, is provided to allow a comparison with other papers.

### 3.2. MRSA

Checkerboard analysis indicated synergistic activity between PRE and Zn (II) against MRSA, with a FIC index of <0.5 (*data not shown*). Figure 2A shows the log reduction of MRSA CFUs with PRE (1 mg/mL) and Zn (II) (0.5 M) alone and in combination, up to 1 h incubation time. Despite negligible changes in MRSA CFUs were identified with PRE alone over the 1 h time course (*p* > 0.05), the combination of PRE with Zn (II) induced significant log reductions in CFUs compared to PRE alone, at all time-points analyzed (*p* < 0.0001–0.001). Although Zn (II) alone also induced significant log reductions in CFUs compared to PRE alone at 20 min (*p* < 0.0001), these responses were to a lower magnitude than with PRE and Zn (II) combined, given the significantly greater log reductions induced by the combinational treatment (*p* < 0.0001). No discernible differences between 0.5 M Zn (II) and the combined treatment were evident at 1 h (*p* > 0.05), as the maximum kill had likely occurred.

Further analyses involving PRE (1 mg/mL) with higher concentrations of Zn (II) (1 M) alone and in combination, up to 1 h incubation time, are shown in Figure 2B. PRE alone again evoked negligible changes in MRSA CFUs over the 1 h time course (*p* > 0.05). The combination of PRE with Zn (II) induced further significant log reductions in CFUs compared to PRE alone, at all time-points analysed (all *p* < 0.0001). Zn (II) alone again induced significant log reductions in CFUs compared to PRE alone, although given the higher concentrations used (1 M), significant Zn (II) effects were shown at 10 min, 20 min, and 1 h, compared to PRE alone (*p* < 0.0001–0.05). However, these responses were more equivalent to those with PRE and Zn (II) combined, as although significantly greater log reductions induced by the combinational treatment at 10 min (*p* < 0.001), Zn (II) alone promoted significantly larger reductions in MRSA CFUs at 20 min (*p* < 0.0001). Again, no obvious differences between 1 M Zn (II) and the combined treatment were evident at 1 h (*p* > 0.05), as the maximum kill had likely occurred. Thus, although such findings implied that PRE and Zn (II) exerted a synergistic effect on MRSA CFUs, this was more apparent at Zn (II) concentrations of 0.5 M, as the synergy became less evident as Zn (II) concentrations increased (1 M).

### 3.3. S. epidermidis

Checkerboard analysis revealed synergistic activity between PRE and Zn (II) against *S. epidermidis*, with a FIC index of <0.5 (*data not shown*). For *S. epidermidis* the concentration range of Zn (II) was expanded to include the lower concentrations of 0.25 M and 0.125 M over maximum incubation time of 1h. As above, negligible changes in *S. epidermidis* CFUs were identified with PRE alone over the 1 h time course (*p* > 0.05). However, the combination of PRE with Zn (II) resulted in significant log reductions in CFUs compared to PRE alone at 1 h with 0.125 M Zn (II) (*p* < 0.0001, Figure 3A), 20 min and 1 h with 0.25 M Zn (II) (both *p* < 0.0001, Figure 3B), 10 min, 20 min, and 1 h with 0.5 M Zn (II) (*p* < 0.0001–0.001, Figure 3C); and all time-points with 1 M Zn (II) (*p* < 0.0001–0.05, Figure 3D). Zn (II) alone again induced significant log reductions in CFUs compared to PRE alone at 1 h with 0.125 M, 0.25 M and 0.5 M Zn (II) (all *p* < 0.0001, Figure 3A–C), although higher Zn (II) concentrations (1 M) promoted significant log reductions in CFUs compared to PRE alone over wider timeframes of 10 min, 20 min, and 1 h (*p* < 0.0001–0.05, Figure 3D). However, these responses were at a lower extent than those induced with PRE and Zn (II) combined, given the significantly greater log reductions induced by the combinational treatment overall, at 1 h with 0.125 M Zn (II) (*p* < 0.0001, Figure 3A), 20 min and 1 h with 0.25 M Zn (II) (*p* < 0.0001–0.01, Figure 3B), 10 min, 20 min, and 1 h with 0.5 M Zn (II) (*p* < 0.0001–0.01, Figure 3C); and at 10 min and 20 min with 1 M Zn (II) (both *p* < 0.0001, Figure 3D). No apparent differences between the 0.5 M and 1 M Zn (II) concentrations alone with the combined treatment were evident at 1 h (*p* > 0.05), as the maximum kill had likely occurred.

### 3.4. E. coli

Checkerboard analysis of PRE and Zn (II) effects on *E. coli* demonstrated that the log reductions in CFUs achieved with PRE (1 mg/mL) alone were more apparent over the 20-min time course, than evident with MRSA and *S. epidermidis* (Figure 4A,B). However, the combination of PRE with 0.5 M Zn (II) induced significantly greater log reductions in CFUs compared to PRE alone, at all time-points analysed (all *p* < 0.0001, Figure 4A). Similarly, despite Zn (II) alone also inducing significant log reductions in CFUs compared to PRE alone at 5 min and 20 min (both *p* < 0.0001), the magnitude of the log reductions in CFUs were again to a lesser extent than with the combinational treatment (all *p* < 0.0001).

Increasing the Zn (II) concentration to 1 M (with PRE 1 mg/mL), produced further, more rapid log reductions in CFUs, than PRE and 0.5 M Zn (II) (Figure 4B). PRE alone induced comparable log reduction to Figure 4A (*p* > 0.05), although the combination of PRE with 1 M Zn (II) induced further significant log reductions in CFUs compared to PRE alone, at all time-points analysed (all *p* < 0.0001). Zn (II) alone also induced significant log reductions in CFUs compared to PRE alone at all time-points (all *p* < 0.0001). However, the combinational treatment was shown to promote much greater log reductions in CFUs, compared to Zn (II) alone, at all time-points (all *p* < 0.0001).

### 3.5. P. aeruginosa

Checkerboard analysis of PRE and Zn (II) effects on *P. aeruginosa* revealed that minor log reduction changes in CFUs were achieved with PRE (1 mg/mL) alone over the 20 min time course (Figure 5A,B). However, the combination of PRE with 0.5 M Zn (II) produced significantly greater log reductions in CFUs compared to PRE alone, at all time-points analysed (all *p* < 0.0001, Figure 5A). Although Zn (II) alone also induced significant log reductions in CFUs compared to PRE alone at 5 min and 10 min (both *p* < 0.0001), the log reductions in CFUs were again much larger with the combinational treatment (all *p* < 0.0001).

Additional increases in Zn (II) concentration (1 M) with PRE (1 mg/mL) also promoted more rapid log reductions in CFUs, than PRE and 0.5 M Zn (II) (Figure 5B). PRE alone induced similar log reduction to Figure 5A (*p* > 0.05), although the combination of PRE with 1 M Zn (II) induced further significant log reductions in CFUs compared to PRE alone at all time-points analysed (all *p* < 0.0001). Zn (II) alone also induced significant log reductions in CFUs compared to PRE alone at all time-points (all *p* < 0.0001). However, the combinational treatment was shown to promote much greater log reductions in CFUs, compared to Zn (II) alone, at all time-points (all *p* < 0.0001).

## 4. Discussion

The main scope of this paper was to use PRE/Zn (II) combinations that had shown synergy against another microbes and establish whether it also has efficacy against the four bacteria in the panel—the aim being to work towards the development of a single product that is broad spectrum in its activity. Combination therapy—i.e., the enhancement of the potency of an antimicrobial agent by the simultaneous administration of a second agent—is of increasing interest in the hunt for new approaches to combat anti-microbial resistance. Here, we use the term ‘synergy’, rather than ‘potentiation’, to describe such modulation, as both PRE and—in particular—Zn (II) show levels of activity against microbes when supplemented individually. However, the data suggests that PRE may in fact potentiate the anti-microbial activity of Zn (II) overall.

PRE is known to possess antimicrobial properties and its combination with metal ions has been shown to result in the significant enhancement of such activities. Potentiation of PRE with Fe (II) was found to produce an 11-log reduction in bacteriophage [17], although the enhancement was temporary and its cessation coincided with solution blackening, as Fe (III) was oxidised to Fe (III). The copper (II) ions have also been proposed as a potentiating agent and have shown to be more effective in combination with PRE than Fe (II) and also Zn (II) [18], although the incubation times were shorter than those reported herein. However, Cu (II) is associated with high toxicity when administered to human cells, which means it use as a novel anti-microbial drug would be limited [22,23]. The Zn (II) ion, which has the benefit of lower toxicity, led to significant synergistic microbicidal activity when combined with PRE against HSV, which was not time-limited [24]. These findings were more recently reflected in the developed synergistic anti-microbial activity of PRE and Zn (II) against the bacterium, *M. luteus* [20].

In order to work towards a single broad-spectrum microbicidal product, higher concentrations were used in order to generally align with the viricidal data reported previously against HSV-1 and HSV-2 (Houston et al., 2017) [19]. Here, we sought to examine such effects against a wide panel of bacteria of a more pathogenic nature: MRSA [25], *S. epidermidis* [26], *E. coli* [27], and *P. aeruginosa* [28]. These species are responsible for a diversity of serious topical and GI infections, as described in the citations. To establish reproducibility, two sets of data were obtained independently by two researchers, using PRE prepared on two separate occasions with different PRE extracts. Results, presented as global means based upon two data sets, show statistically significant synergistic bactericidal activity for all four bacterial species investigated, although different levels of sensitivity to the PRE and Zn (II) combination were apparent. Additionally, higher levels of Zn (II) generally led to synergism occurring more rapidly, possibly as a consequence of the increased thermodynamic activity of the compounds in solution. An earlier paper reported significant log reductions in *S. aureus*, *E. coli*, and *P. aeruginosa* were found when PRE was applied with Cu (II), but not when Zn (II) was used [29].

Mechanistically, the synergistic antimicrobial activity observed between PRE and Zn (II) has yet to be fully elucidated, it is likely that Zn (II) and components of PRE such as punicalagin, are acting independently, albeit cooperatively. It has previously been suggested that PRE may show enhanced activity due to redox cycling of co-administered metal ions, such as Fe (II) and Cu (II), which would increase local levels of reactive oxygen species (ROS) [18]. However, Zn (II) is a stable ion that is not known to undergo such redox processes. Mechanistically, punicalagin has been reported to damage the integrity of bacterial membranes. Electron microscopy observations showed that the cell membrane structures of *Salmonella typhimurium* were damaged after treatment with punicalagin, inducing an increase in the extracellular concentrations of potassium and a release of cell constituents [9]. Although it is likely that, as the predominant tannin, punicalagin would exert a major membrane destabilising effect, other tannins and constituents present in lower proportions may further anti-microbial exert effects. As Zn (II) is also known to be toxic to bacteria at certain concentrations possibly due to blocking Mn (II) uptake [30], it can also be postulated that membrane damage caused by punicalagin would allow greater influx of Zn (II) to toxic levels, due to compromised efflux pumping.

Here we also observed that Gram-negative *E. coli* and *P. aeruginosa* were notably less sensitive than the Gram-positive MRSA and *S. epidermidis*. The cell walls of Gram-positive bacteria are predominantly composed of peptidoglycan, whereas the cell walls of Gram-negative bacteria are more complex, with the presence of a plasma outer membrane located outside of the peptidoglycan layers [31]. This layer presents an additional diffusional barrier for the penetration of polar compounds, such as large hydrolysable tannins in PRE, before the outer membrane bilayer core can be reached [32,33]. Additionally, bacterial cell walls generally possess a negative charge at physiological pH [34], although there is variation between species. In Gram-positive bacteria surface charge is due to the presence of teichoic acids linked to either the peptidoglycan or to the underlying plasma membrane—these teichoic acids are negatively charged because of presence of phosphate in their structure. Gram-negative bacteria have an outer covering of phospholipids and lipopolysaccharides, which impart a strongly negative charge to surface of Gram-negative bacterial cells. However, under more acidic conditions—such as pH 4.5, as used in this work—such charges would be suppressed leading to surface charge neutrality. Similarly, the charge of tannin components of PRE is also suppressed at pH 4.5 relative to pH 7.4 (Figure 6). One immediately apparent effect of such neutralisation is the loss of electrostatic repulsion between the bacterial surface charges and tannins would make interaction between them more favourable, potentially leading to the damage as reported previously [9]. In the absence of functioning efflux pumps the very small cationic Zn (II) would be able to penetrate the bacteria to toxic levels and accelerate cell death.

Taken alongside the data obtained previously for HSV and *M. luteus*, based on these collective findings, there is justification for the development of PRE/Zn (II) as a wide spectrum anti-microbial therapeutic system. This is further supported by the anti-inflammatory properties of PRE/Zn (II) against the local arachidonic acid inflammation pathway [35], antioxidant capacities, and wound healing potential [36].

Topical drug delivery generally involves administering lower doses as they are applied locally to the required site, rather than systemically where effects such as a large volume of distribution must be accounted for. The topical route of administration, such as via a gel or spray, is most appropriate for this system because unlike cases such as co-amoxiclav where the two drugs act independently, the synergistic activity observed with PRE and Zn (II) arises through concentration-dependent co-operative interplay that is unlikely to occur following a systemically administered dose. Furthermore, the levels of Zn (II) used here are high, which precludes them from systemic administration. However, topical cream products containing zinc sulfate are available. Sharquie et al. (2008) used 10% (0.35 M) solutions to examine the effects of zinc sulfate on melasma in a clinical trial—in addition to good clinical responses, no side effects were reported apart from a mild stinging sensation reported in a few patients [37]. Furthermore, the same group later used a 25% (0.87 M) zinc sulphate solution to determine effects on actinic keratosis in 100 patients—again, good clinical responses were observed and the treatment was not associated with side effects apart from mild and transient burning sensation which was encountered in the open lesions of one-third of the patient cohort [38]. The levels of zinc sulfate used in our formulations are similar to these two clinical studies. Moreover, Sharquie et al. (2012) proposed using even higher concentrations of zinc sulfate for future studies [38]. Topical application of zinc in high levels also has a history of use in treating eczemas including contact dermatitis [39].

## 5. Conclusions

The findings in this paper provide further evidence to support the development of a novel, broad-spectrum, anti-microbial product based upon the synergistic microbicidal activity PRE and Zn (II). Such a product would be of value in combating multi-drug, anti-microbial resistance worldwide.

## Figures and Tables

**Figure 1 biomolecules-11-01889-f001:**
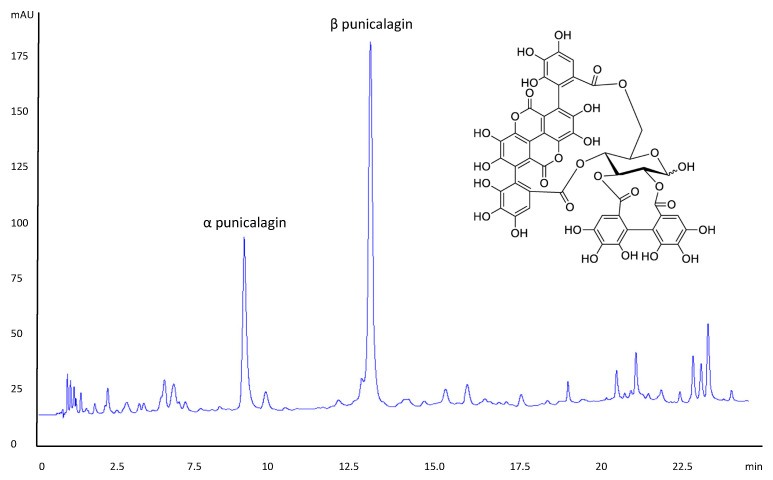
HPLC chromatogram showing PRE elution and highlighting the peaks due to the major ellagitannin punicalagin (α and β anomers), which showed characteristic 1:2 α/β ratio; inset: chemical structure of punicalagin. The level of punicalagin, which is known to be the main active component, was 21.7% (or 1 mg punicalagin contained within 4.61 mg of PRE). NB α and β punicalagin spontaneously exist in this ratio. The full spectrum of components of PRE is described elsewhere.

**Figure 2 biomolecules-11-01889-f002:**
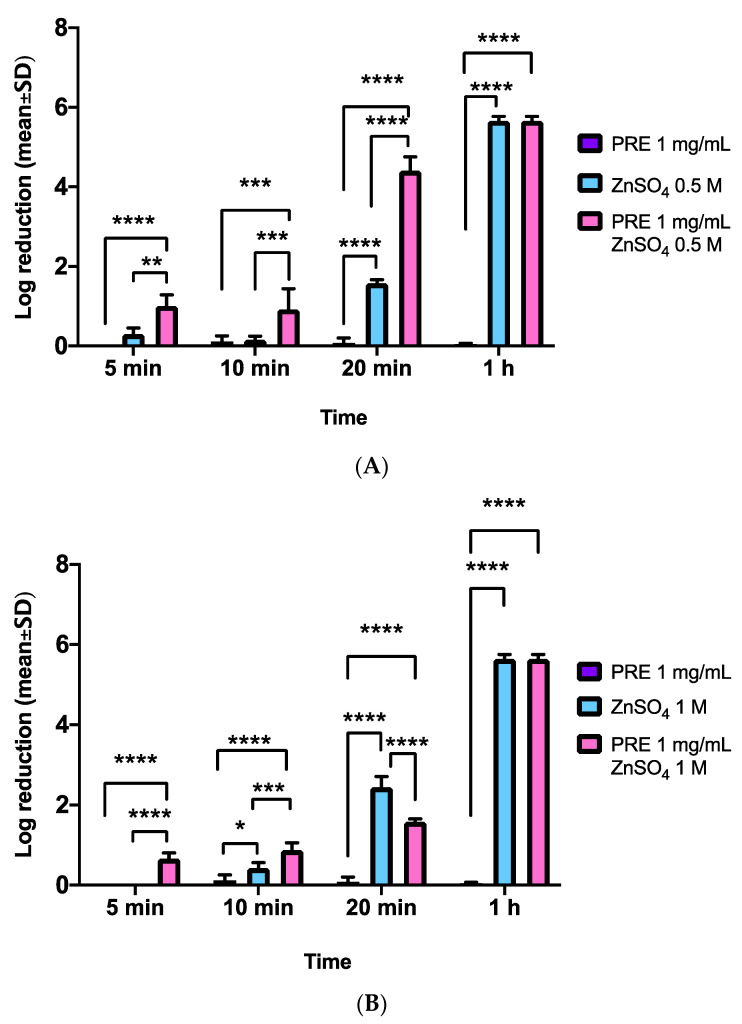
Log_10_ reductions in MRSA CFUs at incubation times of 5 min, 10 min, 20 min, and 1 h (*n* = 9 ± SD). (**A**) PRE (1 mg/mL), and Zn (II) (0.5 M) alone and in combination. (**B**) PRE (1 mg/mL), Zn (II) (1 M) alone, and in combination. Tukey’s multiple comparison post-hoc test show statistically significant differences (* *p* ≤ 0.05, ** *p* ≤ 0.01, *** *p* ≤ 0.001, **** *p* ≤ 0.0001).

**Figure 3 biomolecules-11-01889-f003:**
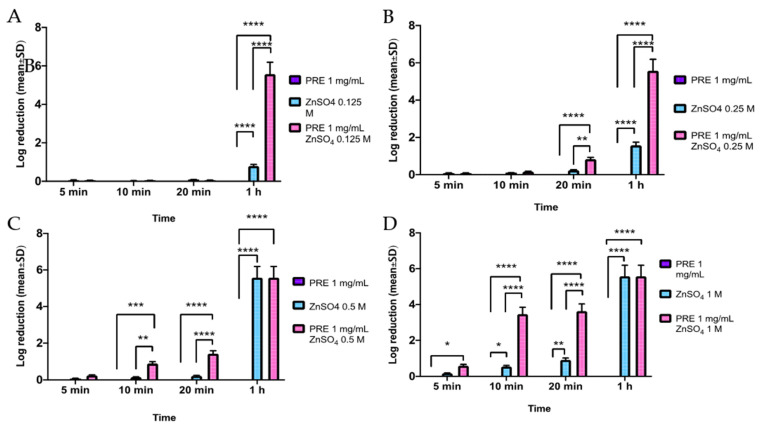
Log_10_ reduction of *S. epidermidis* CFUs at incubation times of 5 min, 10 min, 20 min and 1 h (*n* = 9 ± SD). (**A**) PRE (1 mg/mL) and Zn (II) (0.125 M), alone and in combination. (**B**) PRE (1 mg/mL) and Zn (II) (0.25 M) alone and in combination. (**C**) PRE (1 mg/mL) and Zn (II) (0.5 M) alone and in combination. (**D**) PRE (1 mg/mL) and Zn (II) (1 M) alone and in combination. Tukey’s multiple comparison post-hoc test show statistically significant differences (* *p* ≤ 0.05, ** *p* ≤ 0.01, *** *p* ≤ 0.001, **** *p* ≤ 0.0001).

**Figure 4 biomolecules-11-01889-f004:**
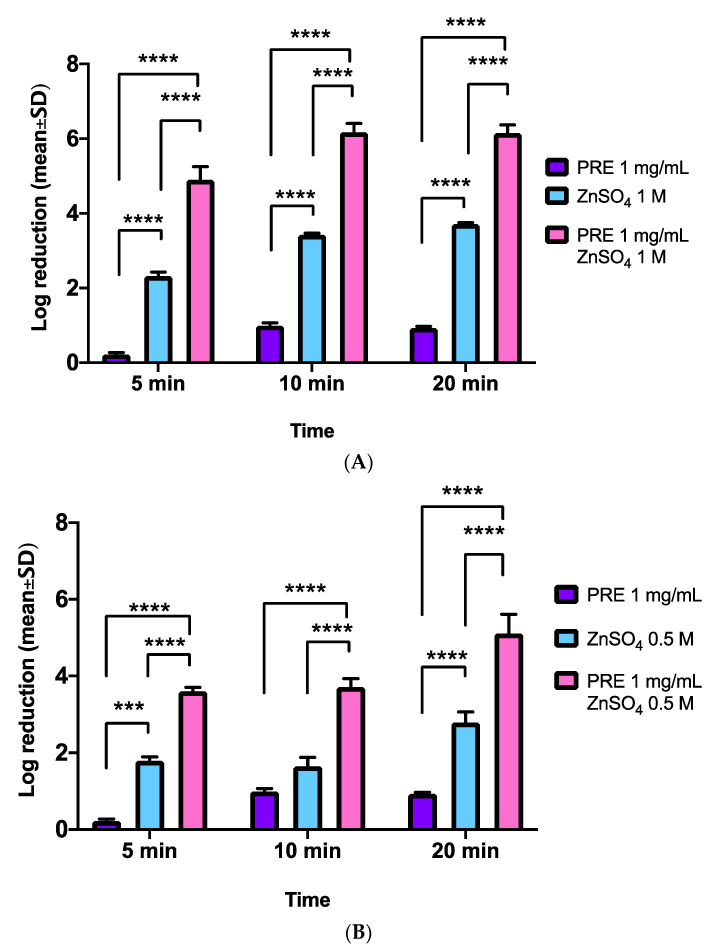
Log_10_ reduction of *E. coli* CFUs at incubation times of 5 min, 10 min and 20 min (*n* = 9 ± SD). (**A**) PRE (1 mg/mL) and Zn (II) (0.5 M) alone and in combination. (**B**) PRE (1 mg/mL) and Zn (II) (1 M) alone and in combination. Tukey’s multiple comparison post-hoc test show statistically significant differences (*** *p* ≤ 0.001, **** *p* ≤ 0.0001).

**Figure 5 biomolecules-11-01889-f005:**
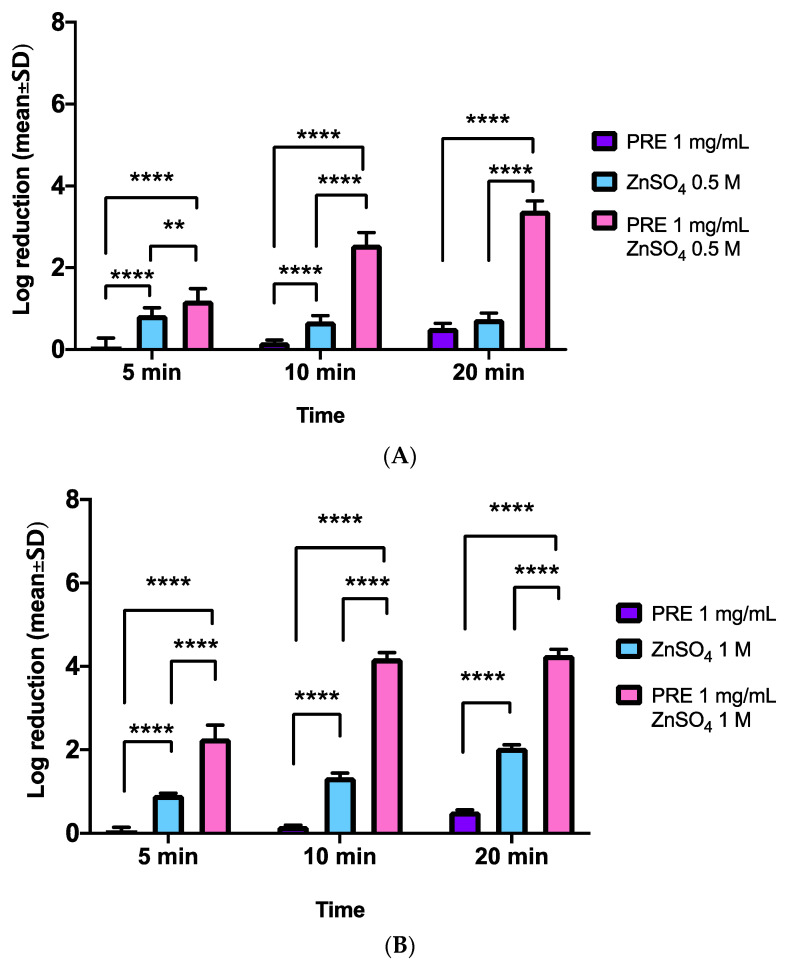
Log_10_ reduction of *P. aeruginosa* CFUs at incubation times of 5 min, 10 min, and 20 min (*n* = 9 ± SD). (**A**) PRE (1 mg/mL) and Zn (II) (0.5 M) alone and in combination. (**B**) PRE (1 mg/mL) and Zn (II) (1 M) alone and in combination. Tukey’s multiple comparison post-hoc test show statistically significant differences (** *p* ≤ 0.01, **** *p* ≤ 0.0001).

**Figure 6 biomolecules-11-01889-f006:**
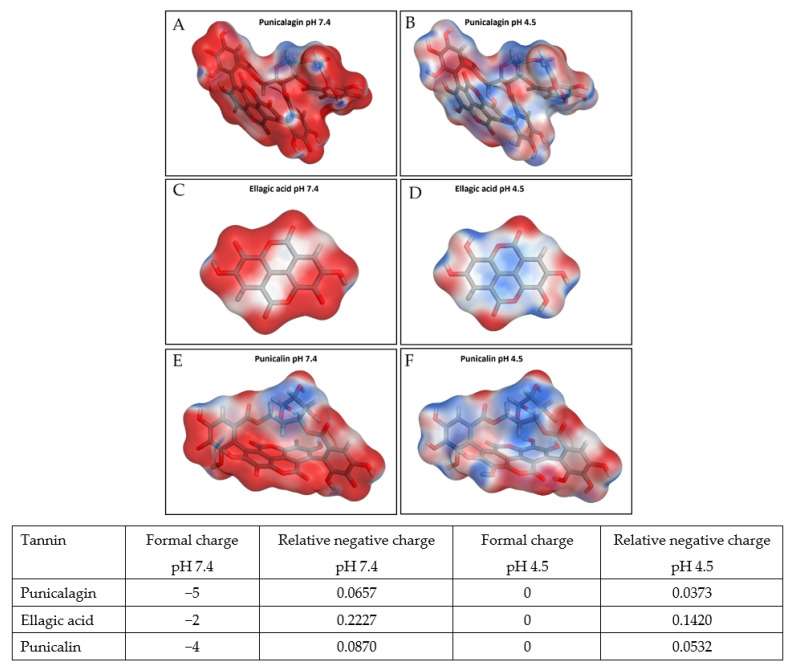
Electrostatic surface maps of punicalagin, ellagic acid, and punicalin at pH 7.4 and pH 4.5. Punicalagin, like other tannins, possesses a negative charge at pH 7.4; however, they are all neutral with almost equal distribution of electrostatic charge positive and negative at pH 4.5. The areas colored in red and blue represent respectively negative and positive regions of the electrostatic potential. Table contains computed charges for punicalagin, ellagic acid, and punicalin, confirming absence of charge at pH 4.5.

**Table 1 biomolecules-11-01889-t001:** Timetable for the HPLC elution of PRE, using a binary gradient elution program. A: methanol + 0.1% trifluoroacetic acid; B: water + 0.1% trifluoroacetic acid.

Time (min)	A. % MeOH + 0.1% TFA	B. % H_2_O + 0.1% TFA
0	5	95
7	10	90
15	20	80
20	40	60
25	60	40
30	5	95

## Data Availability

Not applicable.

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
