# Peer review of "A Time-Kill Assay Study on the Synergistic Bactericidal Activity of Pomegranate Rind Extract and Zn (II) against Methicillin-Resistant Staphylococcus aureus (MRSA), Staphylococcus epidermidis, Escherichia coli, and Pseudomonas aeruginosa"

_biomolecules, 2021, doi:10.3390/biom11121889_

Round 1
Reviewer 1 Report
This study entitled “A Time Kill Assay Study on the Synergistic Bactericidal Activity of Pomegranate Rind Extract and Zn (II) Against Methicillin-Resistant Staphylococcus aureus (MRSA), Staphylococcus epidermidis, Escherichia coli and Pseudomonas aeruginosa” evaluate the synergistic effects of rind pomegranate rind extract (PRE), and Zn (II) ions against the several bacteria and they demonstrated the special efficacy on Gram positive bacteria. However, the authors must be to reduce the abstract since it is is too long and should be more focused on the objective and research documented on the paper.
There has to be some care in the formatting, e.g., species indicated correctly, (e.g. line 229 (E.coli) The title : Staphylococcus; line 136 removed the word dried and Mueller instead Muller.
The references must be corrected according the journal
Author Response
The formatting issues were corrected according to reviewer comment.
Reviewer 2 Report
The manuscript by Alrashidi et al. provides an array of antimicrobial assays on 4 bacteria against pomegranate rind extract and the synergistic effect of Zn. Overall, the combination was shown to have an enhanced effect compared to the single PRE of ZN, although Zn showed a strong activity on its own. Some controls are missing and how this knowledge can be translated into physiological assay would strengthen the manuscript.
Specific queries:
Abstract
Broad spectrum antibiotics are not necessarily a need, in fact they have unfortunately led to fasten the antibiotic resistance levels.
L24: are there 2 or 3 treatments (Zn, PRE, PRE+Zn) ?
L26: Please provide a figure for the synergistic effect.
Introduction
L50-51: please remove this statement.
L58-59: it reads as the regions of the fruit has antimicrobial effect, isn’t the extracts from these regions that contain the compounds with these effects?
M&M
L107: please provide details about the “known concentration”.
L110: sentence is missing a verb
Fig. 1: Please remove the “this is provided to allow […] other papers”
Fig.1: HPLC is not sufficient to characterize the structure/type of the eluted compound, you would need to perform mass spectrometry to confirm this.
L133: was the bacterial suspension under agitation or static? Static incubation could induce anaerobic conditions especially at 60 mins.
L134: what is the neutralizing agent function?
L137 “agar” plate?
L141: was the negative control also using the neutralizing agent? Were positive controls also performed?
Results:
L153: again, the only way to confirm the nature of the HPLC elutes is to use mass spectrometry.
L167: is 0.5M a concentration allowing in vivo testing?
Fig.2: maybe instead of contact, incubation time would be more accurate?
Overall, a positive control would be needed on the figures to gain a sense of the potency.
Discussion:
L314: Gram-negative bacteria expose a lipopolysaccharide layer before the outer membrane bilayer core can be reached.
L334: what is the solubility of the PRE, and in the presence of Zn?
There is a large difference of membrane surface charge between Gram-negative vs positive, could this play a role in the activity of Zn, and ZN+ PRE?
Author Response
Response for Reviewer 2:
- Specific queries for abstract, L24 and L26 was applied.
- Specific queries for introduction, L50-51 and L58-59 were applied.
- Specific queries for M&M, L107, L110, Figure 1, L134, L137 was applied.
- L133: was the bacterial suspension under agitation or static? Static incubation could induce anaerobic conditions especially at 60 mins.
The explanation for the above comment is that the experiment was conducted in a static environment. The reviewer is right about anaerobic condition, but this statement is more apparent for assay performed in small volumes such as 96-well plate-based assays. In our situation, the volume is quite high (1 mL) for emergence of anaerobic condition. Even, anaerobic conditions happened in our experiment, the negative control was kept in the same static condition. Therefore, it could be neglected for this manuscript.
- L141: was the negative control also using the neutralizing agent? Were positive controls also performed?
The necessary details were added in manuscript for first question. For the second question in this comment, positive control group was not performed in this manuscript experimental part.
- L153: again, the only way to confirm the nature of the HPLC elutes is to use mass spectrometry.
To obtain a full spectrum yes, I agree. But to quantitate punicalagin level, HPLC is sufficient.
- L167: is 0.5M a concentration allowing in vivo testing?
This is a high concentration, but is not unprecedented for topical application. High levels of zinc do not have any adverse effects of skin, and are in fact used to treat the inflammatory skin diseases, such as psoriasis
- 2: maybe instead of contact, incubation time would be more accurate?
The contact time was changed to incubation time through the manuscript.
- L314: Gram-negative bacteria expose a lipopolysaccharide layer before the outer membrane bilayer core can be reached.
Hopefully this has been adequately dealt with in the material added to the Discussion
- L334: what is the solubility of the PRE, and in the presence of Zn?
Both Zn and PRE are soluble in media and water used in experiment. According to our knowledge from NMR assay (data not shown in manuscript), we could not find any interaction or complex formation between PRE and Zn. However, in used concentration both compounds exerted a good solubility in this experimental set up.
- There is a large difference of membrane surface charge between Gram-negative vs positive, could this play a role in the activity of Zn, and ZN+ PRE?
This is an excellent point. It does seem that surface charges are an area that require more research, and I agree that it is potentially very important. In fact, charges on the bacteria and tannins should actually be suppressed at the pH used, 4.5. For this paper we have tried to address the point as concisely as we can, and in doing so have added substantially more material including molecular modelling as Fig 6. To help us with this, we recruited the help of Medicinal Chemist, Dr Carmine Varricchio, who had been added as an additional author.
Round 2
Reviewer 2 Report
The manuscript has been nicely revised and can be published.